# Dietary rescue of adult behavioral deficits in the *Fmr1* knockout mouse

**Suzanne O. Nolan**[1]**, Samantha L. Hodges**[2]**, Matthew S. Binder**[1]**, Gregory D. Smith**[2]**, James T. Okoh**[1]**, Taylor S. Jefferson**[1]**, Brianna Escobar**[1]**, Joaquin N. Lugo**[1,2]*****

**1** Department of Psychology and Neuroscience, Baylor University, Waco, Texas, United States of America
**2** Institute of Biomedical Studies, Baylor University, Waco, Texas, United States of America

* joaquin_lugo@baylor.edu

**Data Availability Statement:** The data are available on Figshare: https://doi.org/10.6084/m9.figshare.18321518.v1.

**Funding:** This study received support in the form of funding from the National Institutes of Health

## Abstract

The current study aimed to further address important questions regarding the therapeutic efficacy of omega-3 fatty acids for various behavioral and neuroimmune aspects of the *Fmr1* phenotype. To address these questions, our experimental design utilized two different omega-3 fatty acid administration timepoints, compared to both standard laboratory chow controls ("Standard") and a diet controlling for the increase in fat content ("Control Fat"). In the first paradigm, post-weaning supplementation (after postnatal day 21) with the omega-3 fatty acid diet ("Omega-3") reversed deficits in startle threshold, but not deficits in prepulse inhibition, and the effect on startle threshold was not specific to the Omega-3 diet. However, post-weaning supplementation with both experimental diets also impaired acquisition of a fear response, recall of the fear memory and contextual fear conditioning compared to the Standard diet. The post-weaning Omega-3 diet reduced hippocampal expression of IL-6 and this reduction of IL-6 was significantly associated with diminished performance in the fear conditioning task. In the perinatal experimental paradigm, the Omega-3 diet attenuated hyperactivity and acquisition of a fear response. Additionally, perinatal exposure to the Control Fat diet (similar to a "Western" diet) further diminished nonsocial anxiety in the *Fmr1* knockout. This study provides significant evidence that dietary fatty acids throughout the lifespan can significantly impact the behavioral and neuroimmune phenotype of the *Fmr1* knockout model.

## Introduction

Fragile X syndrome (FXS) is a neurodevelopmental disorder resulting from a trinucleotide (CGG) repeat mutation in the fragile X mental retardation (*FMR1)* gene, which codes for the Fragile X mental retardation protein (FMRP). Mutations in *FMR1* are one of the most prevalent genetic contributors to inherited intellectual disability [1] and Autism spectrum disorder (ASD) [2]. In clinical studies, individuals with FXS display a myriad of behavioral abnormalities, including significant cognitive dysfunction, autistic behaviors, and hyperactivity [3]. These phenotypes are often mirrored in the *Fmr1* knockout mouse, which demonstrates significant learning impairments, hyperactivity, social impairments, and increased repetitive

(NS088776, awarded to JL) The funders had no role in study design, data collection and analysis, decision to publish, or preparation of the manuscript.

**Competing interests:** The authors have declared that no competing interests exist.

behavior, among other behavioral phenotypes [4–6]. Alongside these behavioral phenotypes, recent clinical investigations have also demonstrated significant evidence for the role of cytokine signaling in individuals with FXS [7, 8]. Previous studies have shown evidence of aberrant proinflammatory signaling as a key feature in both FXS and its comorbid condition, ASD, in both clinical populations and animal models [7–10].

Omega-3 fatty acids, which serve as potent anti-inflammatory dietary compounds, have been investigated for their potential to alleviate behavioral symptoms in clinical populations of both ASD [11–15] and Rett syndrome [16]. Treatment with omega-3 fatty acids incorporates several potential mechanisms that may be relevant to the *Fmr1* knockout. First, eicosapentaenoic acid (EPA) and docosahexaenoic acid (DHA), two of the primary omega-3 fatty acids, are significant components of the neuronal membrane, and their composition in this membrane regulates functioning of key transmembrane receptors and other proteins [17]. Previous work has demonstrated that even one membrane substitution of a single-bonded polyunsaturated fatty acid (PUFA) for a double-bonded PUFA exerts changes in membrane properties that in turn alter constitutive activity of integral membrane proteins, such as rhodopsin [18]. Mechanistically, increasing the bioavailability of long chain PUFAs may also enable important anti-inflammatory mechanisms [19]. These may be accomplished through a variety of methods, including interactions with G-protein coupled receptors or downstream production of small anti-inflammatory compounds [20]. There is also significant evidence for an influence of DHA on markers of synaptogenesis and expression of glutamate receptors [21]. Further, limited preclinical evidence finds that post-weaning treatment with omega-3 fatty acids resulted in improvements in novelty exploration, social interaction, and object recognition in association with reductions in neuroinflammatory signaling markers in the adult *Fmr1* mouse model [10].

While the aforementioned evidence supports a role of omega-3s as a potential therapeutic for the *Fmr1* knockout phenotype, it is unknown if administration of this intervention at an earlier timepoint would result in differential effects on the behavioral and neuroinflammatory phenotypes studied in the present model. Recent work has shown promising effects of perinatal administration of omega-3s in *Fmr1* mutant rats, supporting its potential for rescuing behaviors in this model [22]. Mechanistically, accumulation of fatty acids is fundamental for the formation of neuronal membranes, particularly early in development [23]. Studies of fatty acid incorporation demonstrate that essential long chain PUFAs, such as DHA and EPA, are not absorbed and incorporated appropriately during development in children with ASD [24]. Moreover, increased bioavailability of fatty acids during early development can induce discernable phenotypic changes as early as the weaning point [25]. While no studies have examined the therapeutic potential of perinatal administration in the *Fmr1* model, previous studies have supported the potential for prenatal omega-3 fatty acids in the reversal of autistic-like deficits in other ASD rodent models [26–30]. Moreover, interventions during this timeframe provide a higher likelihood for maximal impact, given that administration of omega-3 fatty acids after diagnosis in clinical studies may be too late to improve outcomes [31].

The current study aimed to address questions regarding the therapeutic efficacy of omega-3 fatty acids for key aspects of the *Fmr1* phenotype, such as whether any behavioral effects might be mediated through changes in inflammatory gene expression, whether timing matters, and how previous effects compare when referenced to a standard dietary control. It should be noted that the previous work utilized a dietary control to compare their behavioral findings against, which represents an increase in fat content from typical laboratory chows [10]. The behavioral measures were selected as they most appropriately reflect the FXS clinical phenotype [32], and were based on prior findings [5]. In the first experiment, we exposed male *Fmr1* knockouts to dietary manipulations in the post-weaning period and examined various aspects

of adult behavior and cytokine signaling markers. In the second experiment, we administered similar dietary manipulations during the perinatal period to investigate how perinatal dietary manipulations prior to weaning will influence these same behaviors and cytokine signaling markers during adulthood in male *Fmr1* KO mice.

## Materials and methods

### Animals

All procedures were performed in accordance with *Baylor University Institutional Care and Use Committee* and the *Guide for the Care and Use of Laboratory Animals of the National Institutes of Health*. The protocol was approved by the Baylor University Institutional Animal Care and Use Committee (Protocol number: 744037–3). Male *Fmr1*$^{+/+}$ and female *Fmr1*$^{+/-}$ *FVB.129P2-Pde6b+Tyrc-ch Fmr1tm1Cgr/J* (Jackson Labs Stock No: 004624) mice originally from Jackson Labs were housed at Baylor University and bred to produce male wildtype (WT) and *Fmr1* knockout (KO) offspring. The colony was maintained on a 12-hour light/dark cycle (lights on at 7 am). On postnatal day (PD) 7, pups were separated from parents and a toe-clipping identification system was used to identify animals and the tissue was sent out for genotyping and preserved in 70% ethanol (Mouse Genotype, Escondido, CA, USA). Animals were housed in the Special Research Unit at the Baylor Science Building. All tested animals and their breeders had access to food and water *ad libitum*.

Only male mice were used for the present studies, as previous work from our lab did not find strong evidence of a phenotype in female mutants in many of the tasks included here [5]. However, other work from our lab has found evidence of robust sex-specific effects in ultrasonic vocalization behavior in this model [33, 34] and has explored the impact of these dietary manipulations on this behavior in other studies [35].

### Experimental design

The studies were split into two main experimental paradigms (Fig 1). In the first experimental paradigm, male offspring were maintained on standard laboratory chow until weaning on PD21 (Harlan Teklab 2020X formulation). At PD21, animals were randomly assigned to one of three diet conditions: "Standard", "Omega-3" and "Control Fat". The latter two diets had identical lipid content (50g/kg). The Omega-3 condition received a diet enriched with fish oil and dyed with red food coloring to distinguish it from the control condition (Teklad Custom Diet TD.150384). To control for potential effects due to simply increasing fat content, the Control Fat diet contained olive and palm oils with no dyes (Teklad Custom Diet TD.1500385) (See Table 1 for Ingredients List). The dietary composition was based on levels previously used in this model [10]. For further information of the types of fatty acid chains included, see Table 2. Mixed-genotype littermates were housed in groups no larger than 5 animals, with each cage receiving the same diet assignment. No culling was performed in either feeding paradigm. Based on the methodologies of prior studies, animals were maintained on this diet and tested in the behavioral paradigm starting at PD90 [10]. Animals were maintained on the diet throughout the testing paradigm. We chose PD60 to start testing after the perinatal treatment because it was the earliest day that the mice would be considered a young adult. We chose PD90 for the postnatal treatment so the mice could have a longer exposure to the high omega-3 diet. The two time periods were chosen to maximize the effect of the diet. The behavior tests were ordered in a way that increased in invasiveness, to minimize the effect of training history on subsequent behavioral tests [36]. The final sample sizes were as follows: standard diet WT (n = 11), standard diet KO (n = 10), control fat WT (n = 17), control fat KO (n = 15), omega-3 WT (n = 19), omega-3 KO (n = 16). Due to equipment errors, data was not available for

# Post-weaning Paradigm

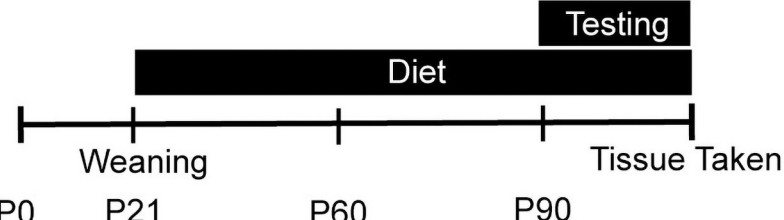

# Perinatal Paradigm

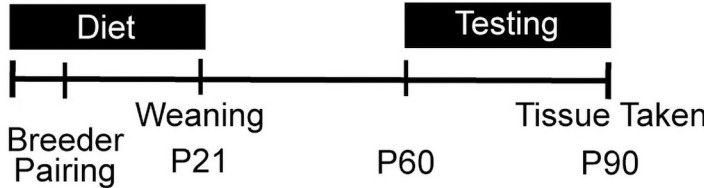

**Fig 1. Overview of the two feeding paradigms.** For the postnatal paradigm, the pups were weaned onto one of three dietary manipulations and maintained on this diet through the conclusion of the behavioral testing paradigm. For the perinatal paradigm, breeders were placed on the dietary manipulations for one week prior to pairing. Breeding pairs and litters were maintained on this diet until weaning on PD21, at which time male pups were weaned onto standard diet. All animals in this paradigm received standard diet from PD21 through the conclusion of testing.

**Table 1. Diet ingredients for post-weaning paradigm.**

| Ingredients, g/kg | Control for EPA and DHA Diet (Control Fat) | Omega-3 EPA and DHA Diet (Omega-3 Diet) |
|---|---|---|
| Casein | 200.0 | 200.0 |
| L-Cystine | 3.0 | 3.0 |
| Corn Starch | 347.472 | 347.472 |
| Maltodextrin | 132.0 | 132.0 |
| Sucrose | 100.0 | 100.0 |
| Canola Oil | 13.0 | 10.0 |
| Olive Oil | 65.0 | 20.0 |
| Palm Oil | 42.0 | 10.0 |
| Fish Oil | - | 80.0 |
| Cellulose | 50.0 | 50.0 |
| Mineral Mix, AIN-93G | 35.0 | 35.0 |
| Vitamin Mix, AIN-93 | 10.0 | 10.0 |
| Choline Bitartrate | 2.5 | 2.5 |
| TBHQ, antioxidant | 0.028 | 0.028 |
| Total | 1000.0 | 1000.0 |

This table contains ingredients for Teklad Custom Diets TD.1500384 and TD.1500385.

**Table 2. Fatty acid breakdown.**

|  | Fatty Acids (% of Total)<br>Control Fat Diet | Omega-3 Diet |
|---|---|---|
| SFA | 27.9 | 26.8 |
| MUFA | 59.8 | 36.8 |
| PUFA | 12.2 | 36.4 |
| 14:00 | 0.4 | 5.7 |
| 16:00 | 23.6 | 16.5 |
| 16:01 | 0.7 | 7.9 |
| 18:00 | 3.2 | 2.8 |
| 18:01 | 59.0 | 26.8 |
| 18:02 | 10.7 | 5.2 |
| 18:03 | 1.5 | 2.2 |
| 18:04 | 0.0 | 2.3 |
| 20:5 (EPA) | 0.0 | 10.7 |
| 22:6 (DHA) | 0.0 | 7.2 |
| n-3 | 1.5 | 22.6 |
| n-6 | 10.7 | 6.7 |

Fatty acid breakdown for Teklad Custom Diets TD.1500384 and TD.1500385.

certainanimals for some behaviors, and final sample sizes for each behavioral test in this paradigm are delineated in Table 3.

In the second experimental paradigm, breeders were placed on one of the three experimental diets one week prior to pairing (Fig 1). Both parents and offspring were maintained on the assigned diet throughout pregnancy, parturition and lactation. Many aspects of the experimental diets were the same from paradigm 1, however, there were some additional nutrients added for breeding suitability (for a list of ingredients, see Table 4). Animals were then weaned onto standard laboratory chow on PD21 and housed with mixed genotype littermates in groups of no larger than 5 mice. Male animals were then tested in the behavioral battery beginning at PD60. The final sample sizes were as follows: standard diet WT (n = 24), standard diet KO (n = 12), control fat WT (n = 16), control fat KO (n = 16), omega-3 WT (n = 14), omega-3 KO (n = 14). Due to equipment errors, some data was not available for all animals in some behaviors, and final sample sizes for each behavioral test for this paradigm are delineated in Table 5. Any additional data can be found in the S1 File.

## Elevated plus maze

The elevated plus maze task was performed to evaluate changes in baseline anxiety levels [37, 38]. The testing room was lit by LED dimmable lamps (30 lux in the open arm) and the

**Table 3. Sample sizes for post-weaning paradigm.**

| Behavioral Test/Measure | Standard | | Control Fat | | Omega-3 | | Total |
|---|---|---|---|---|---|---|---|
|  | WT | KO | WT | KO | WT | KO |  |
| Elevated Plus Maze | 9 | 10 | 14 | 16 | 17 | 16 | 82 |
| Sensorimotor Gating Paradigms | 11 | 9 | 16 | 16 | 19 | 16 | 87 |
| Delay Fear Conditioning | 11 | 10 | 15 | 15 | 19 | 16 | 86 |
| PCR | 6 | 5 | 6 | 6 | 6 | 6 | 35 |

Sample sizes for each behavioral test following exclusion for technical or scoring issues.

**Table 4. Diet ingredients for perinatal paradigm.**

| Ingredients, g/kg | Control for EPA and DHA Diet (Control Fat) | Omega-3 EPA and DHA Diet (Omega-3 Diet) |
|---|---|---|
| Casein | 200.0 | 200.0 |
| L-Cystine | 3.0 | 3.0 |
| Corn Starch | 341.60 | 341.60 |
| Maltodextrin | 132.0 | 132.0 |
| Sucrose | 100.0 | 100.0 |
| Canola Oil | 13.0 | 10.0 |
| Olive Oil | 65.0 | 20.0 |
| Palm Oil | 42.0 | 10.0 |
| Fish Oil | - | 80.0 |
| Cellulose | 50.0 | 50.0 |
| Mineral Mix AIN-93G | 35.0 | 35.0 |
| Vitamin Mix AIN-93 | 10.0 | 10.0 |
| Choline Bitartrate | 2.5 | 2.5 |
| Calcium Phosphate, dibasic | 3.1 | 3.1 |
| Calcium Carbonate | 1.0 | 1.0 |
| Magnesium Oxide | 0.154 | 0.154 |
| Cupric Carbonate | 0.0038 | 0.0038 |
| Ferric Citrate | 0.2352 | 0.2352 |
| Sodium Selenite (0.0455% in sucrose) | 1.25 | 1.25 |
| Vitamin $K_1$, phylloquinone | 0.0003 | 0.0003 |
| Vitamin $B_{12}$ (0.1% in mannitol) | 0.025 | 0.025 |
| TBHQ, antioxidant | 0.028 | 0.028 |
| Total | 1000.0 | 1000.0 |

Ingredients for Teklad Custom Diets TD.160486 and TD.160487.

background noise remained at 60 dB. The testing arena consisted of four arms (30 x 5 cm) and a center platform (5 x 5 cm) positioned approximately 40 cm above the floor. Two opposing arms were enclosed by acrylic walls. Animals were recorded for 10 minutes and their movement was assessed via Ethovision XT video tracking software (Noldus, Netherlands). This program scored the frequency and duration of visits to the various arms and center platform, as well as distance moved during the entire task and average velocity over the testing window. The percent of time in the open arms was calculated by taking the total duration of time spent across all visits to the arms as a function of the total testing time (10 minutes). The testing

**Table 5. Sample sizes for perinatal paradigm.**

| Behavioral Test/Measure | Standard | | Control Fat | | Omega-3 | | Total |
|---|---|---|---|---|---|---|---|
| | WT | KO | WT | KO | WT | KO | |
| Elevated Plus Maze | 22 | 10 | 17 | 13 | 14 | 14 | 90 |
| Sensorimotor Gating Paradigms | 23 | 9 | 15 | 16 | 14 | 13 | 90 |
| Delay Fear Conditioning | 24 | 12 | 16 | 16 | 14 | 14 | 96 |
| PCR | 6 | 6 | 6 | 6 | 6 | 6 | 36 |

Sample sizes for each behavioral test following exclusion for technical or scoring issues.

apparatus was cleaned thoroughly with 30% isopropyl alcohol solution before and after each subject was tested. Experimenters were not present during the testing window. Final sample sizes for each paradigm are delineated in Tables 3 and 5.

## Sensorimotor gating assessment

To determine changes in sensorimotor gating abilities in the *Fmr1* KO mice, the sensorimotor gating assessment paradigm was implemented as previously described [39]. Briefly, the apparatus consisted of an acrylic hollow constraint tube, with varying degrees of restraint availability. This apparatus was mounted on a platform equipped to transduce startle response amplitude through the SR-Lab System (San Diego Instruments, San Diego, CA, USA). This paradigm consisted of three separate testing days. During all testing sessions, background levels were maintained at 68 dB. On the first day, the animals underwent a habituation session. During this, they had a 5-minute acclimation session, followed by 80 startle stimuli delivered at a fixed interval (15 s). The startle stimulus was a 40 ms, 120 dB noise burst, with a rise/fall time of less than 1 ms.

The next day, prepulse inhibition was assessed. Following a 5-minute habituation phase, animals received 20 presentations of a 40 ms, 120 dB stimulus. They were then presented with the prepulse phase of the trial, consisting of 90 trials. The first three trial types consisted of the 20 ms prepulse stimulus at three different decibel levels (70, 75 and 80 dB). The second three trial types then consisted of these three prepulse stimuli paired with the original startle stimulus. The prepulse stimulus was then preceded by the startle stimulus by 100 ms. These trials were organized randomly and spaced by a 15 s inter-trial interval.

One week following the prepulse session, the startle threshold session was conducted. Following the initial 5-minute habituation period, mice were presented with 99 trials of 11 trial types: no stimulus, and ten startle stimuli ranging from 75–120 dB at five dB intervals. These startle stimuli lasted 40 ms with a rise/fall of less than 1 ms. The order of these trials was pseudorandomized, such that 11 trials were presented as a block in a different order each time. Experimenters were not present during the testing window. Final sample sizes for both paradigms are delineated in Tables 3 and 5.

## Delay fear conditioning

The delay fear conditioning paradigm was conducted, as it has been shown to be selective for amygdala-based fear memories [40]. On the first day of testing, following a 2 minute baseline period, the animals received 2 pairings of a 30-second 80-dB white noise stimulus (designated the CS) followed by a 0.7 mA shock stimulus (designated the US). Following each pairing, there was a 120 second inter-trial interval (ITI). The session lasted approximately 334 seconds.

On the second day, there were two testing sessions. During the first, the animal was placed in the shock-associated context and allowed to move freely for 300 seconds to evaluate freezing in the original context (contextual freezing behavior). Following a two-hour rest, the animal was placed in a new context for 360 seconds. The context was altered by the following manipulations: a clear acrylic square placed over the shock grid (novel tactile context), the shape of the arena altered by the insertion of an acrylic panel, shredded paper towels in the transfer cage and 1 mL of pure vanilla extract placed beneath the floor (novel olfactory context). During the first 3 minutes, we measured the freezing behavior of the mouse in a new environment. In the second 3 minutes, the CS was presented continuously for 3 minutes. During all sessions, freezing behavior was measured by an automated software (Colbourn Instruments, Allentown, PA, USA). Experimenters were not present during the testing window. Final sample sizes for this test for each experimental paradigm are delineated in Tables 3 and 5.

## Quantitative real-time polymerase chain reaction

Following completion of behavioral testing, the brain was removed and rinsed in 1X phosphate buffer (PBS) solution (for exact timing, please refer to Fig 1). Using previously described methods [41], hippocampi were then rapidly dissected from each hemisphere and quickly rinsed in ice cold 1X PBS, before being frozen on dry ice and stored in microcentrifuge tubes at -80 ºC until processing. The left hippocampus was used for all PCR assays. Total RNA was isolated from samples according to protocols from the RNeasy kit Qiagen, Hilden, Germany). Subsequently, concentration and purity of isolated samples were measured using a NanoDrop ND-1000 Spectrophotometer (Thermo Scientific, NanoDrop Products, Wilmington, DE). Using the High-Capacity cDNA Reverse Transcription Kit (Applied Biosystems, Carlsbad, CA), extracted RNA was then reverse transcribed into complementary DNA. Using a QuantStudio 5 Real-Time PCR System (Applied Biosystems, Carlsbad, CA), mRNA expression was then determined by quantitative real-time polymerase chain reaction (qRT-PCR) using Taq-man probe and primer chemistry. Reactions were performed in triplicate in a 384 well plate for each sample, using the endogenous control gene (β-actin) for normalization. The expression levels of each target gene, BDNF, IL-6, TNF-α and IL-1β, was calculated by normalizing the quantified mRNA amount to β–actin using the $2^{-\Delta\Delta Ct}$ method of quantitation. Relative gene expression of each group was calculated in relation to the Standard Diet WT group and used to test significance between groups.

## Statistical analyses

All data were analyzed using GraphPad Software 7.0 (San Diego, CA) or IBM SPSS Statistics 23 (Aramonk, NY). Results were evaluated using a 2 x 3 (Genotype [WT, KO] x Diet [Standard, Control Fat, Omega-3]) analysis of variance (ANOVA) on each dependent variable for the specific test. Any tests that involved repeated measures were analyzed using a within- subjects factor, by repeated measures ANOVA (specified in the appropriate results section). Significant within-subjects interactions were followed up with individual one-way ANOVAs at each repeated measure. Significant interactions of genotype and diet were followed up with the use of a unique identifier for all groups (i.e. "Standard WT") and subsequent analysis. If there were multiple significant main effects, multiple comparisons were conducted using Fisher's LSD comparisons (Standard WT vs Standard KO, Standard WT vs Control Fat KO, Standard WT vs Omega-3 KO). For PCR, following a significant main effect, multiple comparisons were again conducted using Fisher's LSD comparisons. All results have been summarized for convenience, and all statistical results including correlations between cytokine expression levels and behavioral measures, which were conducted using Pearson's correlation statistic, are summarized in the S1 File. For all inferential statistics, the level of significance remained at $p < 0.05$.

## Results

### Perinatal but not post-weaning supplementation with a high omega-3 diet attenuated hyperactivity in the elevated plus maze in the *Fmr1* knockout mouse, with no effect on anxiety

Hyperactivity is a significant clinical component of FXS [42, 43], and has been well-described in the *Fmr1* KO [5, 44–46]. Given the potential impact of hyperactivity on the findings of subsequent behavioral testing and the consistent appearance in the *Fmr1* KO model, we assessed the impact of high omega-3 fatty acids on activity levels in the elevated plus maze. For the post-weaning paradigm, in the elevated plus maze, loss of *Fmr1* did not influence distance moved, $F(1, 76) = 1.57$, $p = 0.21$ (Fig 2A). Post-weaning diet also did not influence distance

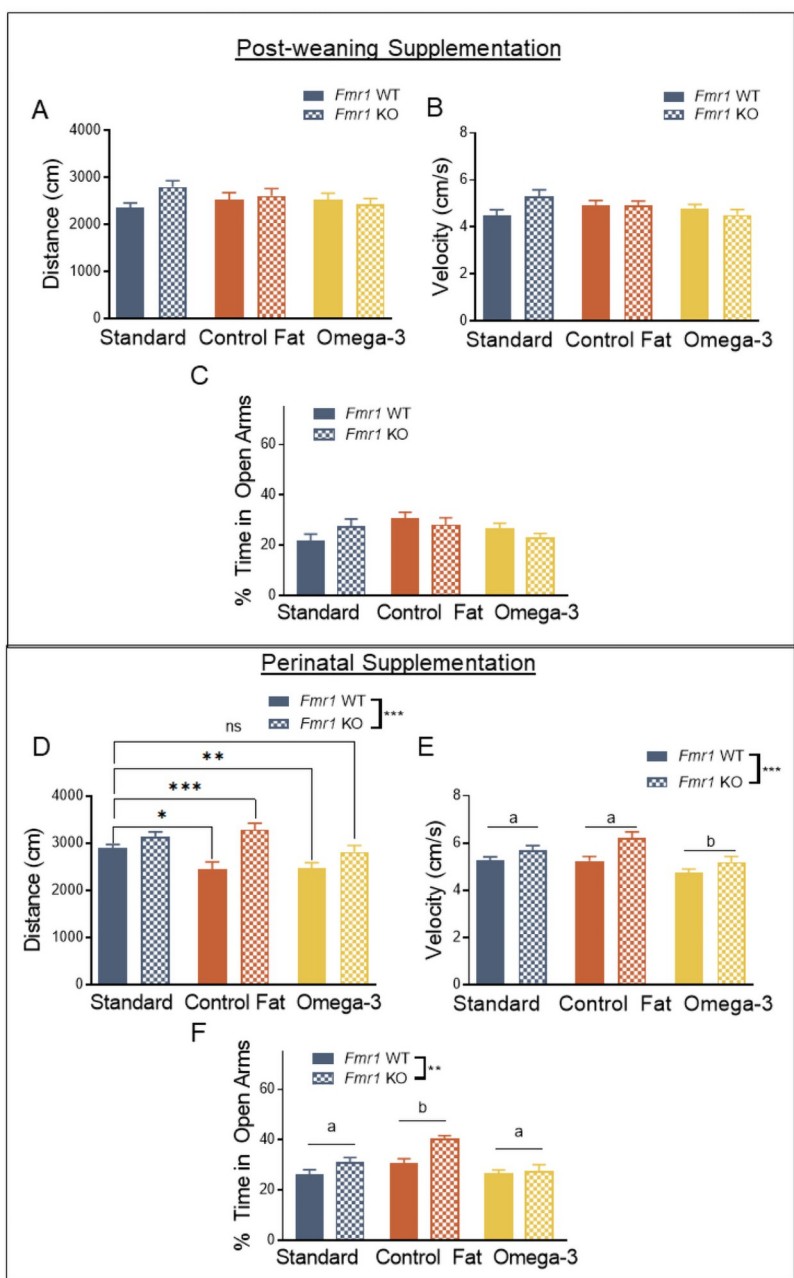

**Fig 2. Perinatal supplementation with a high omega-3 diet attenuated hyperactivity in the elevated plus maze in the *Fmr1* knockout mouse, with no effect on anxiety.** A. Exposure to omega-3 fatty acids had no effect in the post-weaning paradigm as measured by distance moved in the elevated plus maze. B. A similar pattern was detected for velocity in the elevated plus maze. C. No effect of the experimental diets was seen in the post-weaning paradigm. D. Perinatal exposure to omega-3 fatty acids reversed *Fmr1* knockout induced hyperactivity in the elevated plus maze, as measured by distance moved. E. Perinatal exposure to omega-3s also reversed hyperactivity as measured by velocity in the elevated plus maze. F. Loss of *Fmr1* was associated with diminished anxiety in the elevated plus maze, though this effect was not ameliorated by exposure to omega-3 fatty acids. Rather, exposure to the control fat perinatal diet exacerbated this genotype effect. Data are expressed as mean ± SEM.* = $p < 0.05$, ** = $p < 0.01$, *** = $p < 0.001$ ANOVA. A designation of "b" indicates that this group differed from the "a" comparison group at the level of $p < 0.05$ in post-hoc analyses.

moved, $F(2, 76) = 0.32$, $p = 0.74$. Moreover, there was no significant interaction of genotype and this diet, $F(2, 76) = 2.75$, $p = 0.07$.

A similar pattern was detected for velocity. Alone, loss of *Fmr1* did not significantly increase velocity, $F(1, 76) = 0.88$, $p = 0.35$ (Fig 2B) and post-weaning diet did not significantly influence velocity on its own, $F(2, 76) = 0.99$, $p = 0.38$. Moreover, there was no interaction between genotype and diet, $F(2, 76) = 2.96$, $p = 0.06$.

Previous studies conducted in this model have indicated that non social anxiety is significantly reduced following loss of *Fmr1* [4]. To examine the influence of these dietary paradigms on anxiety, we assessed animals in the elevated plus maze, and recorded the percentage of time spent in open arms of the maze. For the post-weaning paradigm, loss of *Fmr1* did not impact anxiety in this task, $F(1, 76) = 0.003$, $p = 0.96$, and diet was also not a significant factor, $F(2, 76) = 2.03$, $p = 0.14$ (Fig 2C). These variables also did not significantly interact, $F(2, 76) = 1.54$, $p = 0.22$. The results suggest that the reduced anxiety characteristic of the *Fmr1* KO is not present when tested around PD90 in this model, unlike previous studies conducted at an earlier time points [5]. Furthermore, neither post-weaning dietary fatty acid manipulation affected the anxiety phenotype of either WT or KO animals.

For the perinatal paradigm, in the elevated plus maze, loss of *Fmr1* resulted in hyperactivity when measuring distance moved, $F(1, 84) = 21.04$, $p = 0.001$ (Fig 2D). However, the perinatal diet did significantly influence distance moved, $F(2, 84) = 4.26$, $p = 0.02$. Post-hoc multiple comparisons with LSD indicated that animals, across genotypes, receiving omega-3 fatty acids had reduced distance moved and velocity when compared to both standard and control fat conditions, $p < 0.05$ (Fig 2D). A significant interaction was detected for distance moved, $F(2, 84) = 3.35$, $p = 0.04$, and a follow-up analysis with unique group identifiers (i.e. Standard Diet WT) was conducted according to our a priori hypotheses. Interestingly, the perinatal control fat diet significantly exacerbated hyperactivity in the KO (Control Fat KO vs Standard WT, $p = 0.02$). However, the perinatal control fat diet reduced hyperactivity in the wildtypes (Control Fat WT vs Standard WT, $p = 0.004$). This was similarly true for the omega-3 WT diet (Omega-3 WT vs Standard WT, $p = 0.01$). Further, the omega-3 KO group expressed levels of activity similar to the standard WT group, $p = 0.62$ (Fig 2D), suggesting this rescued the phenotype seen in the Standard KO.

The results for velocity were similar. Overall, loss of *Fmr1* increased velocity, $F(1, 84) = 14.41$, $p = 0.001$ (Fig 2E). While no interaction was detected, $F(2, 84) = 1.21$, $p = 0.30$, exposure to the different perinatal dietary manipulations did indeed affect velocity, $F(2, 84) = 7.52$, $p = 0.001$. Post-hoc LSD comparisons indicated that the Omega-3 Diet ("b") condition significantly lowered velocity, relative to both the Standard Diet ("a") (p = 0.29), as well as the Control Fat ("a") diet (p = 0.001), suggesting that this aspect of hyperactivity was also rescued.

For the perinatal paradigm, loss of *Fmr1* resulted in decreased anxiety, $F(1, 84) = 7.91$, $p = 0.01$, as suggested by increased percentage of time spent in the open arms (Fig 2F). Increased time spent in the open arm in the *Fmr1* KO was significantly exaggerated by exposure to the control fat perinatal diet, $F(2, 84) = 8.01$, $p = 0.001$. Post-hoc analyses indicated that the control fat perinatal diet ("b") significantly increased the proportion of time spent in the open arms compared to both standard ("a") and omega-3 perinatal diets ("a"), at the level of $p < 0.05$. However, the standard perinatal diet and omega-3 conditions produced similar effects (Fig 2F). Genotype and perinatal diet also did not significantly interact, $F(2, 84) = 2.10$, $p = 0.13$. These results suggest that the omega-3 fatty acid perinatal diet did not improve altered anxiety characteristic of the *Fmr1* model. Moreover, the control fat perinatal diet, similar to a typical Western diet, further exacerbated this reduced anxiety phenotype.

Overall, similar to the post-weaning paradigm, exposure to omega-3 fatty acids, but not other types of fatty acids, attenuates hyperactivity in the adult *Fmr1* KO measured via the elevated plus maze.

## Post-weaning exposure to high fat diets reversed startle threshold, while neither paradigm reversed PPI deficits

To further examine the potential therapeutic efficacy of omega-3 fatty acids, we examined this behavior in a three-day paradigm similar to previously published methodology [39]. For the post-weaning paradigm, results for habituation to the startle stimulus (data found in S1 File) indicated no impact of loss of *Fmr1* on the ability to habituate to the chamber, $F(1, 81) = 0.001$, $p = 0.97$, and this did not vary across the testing window, $F(7, 567) = 0.76$, $p = 0.62$. Post-weaning diet also did not impact habituation, $F(2, 81) = 0.72$, $p = 0.49$, and this effect did not change over time, $F(14, 567) = 0.89$, $p = 0.57$. The interaction of the different levels of genotype and post-weaning diet were not significant, both overall, $F_t(2, 81) = 1.91$, $p = 0.16$, or across time, $F(14, 567) = 0.52$, $p = 0.92$. Overall, this indicated that neither post-weaning diet nor genotype had a significant impact on habituation to the startle stimulus.

As expected, loss of *Fmr1* resulted in overall exaggerated levels of prepulse inhibition (measured on Day 2), $F(1, 81) = 6.15$, $p = 0.02$ (Fig 3A). This effect was consistent across decibel levels, $F(2, 162) = 2.57$, $p = 0.08$. Because this effect was consistent across the testing window, data

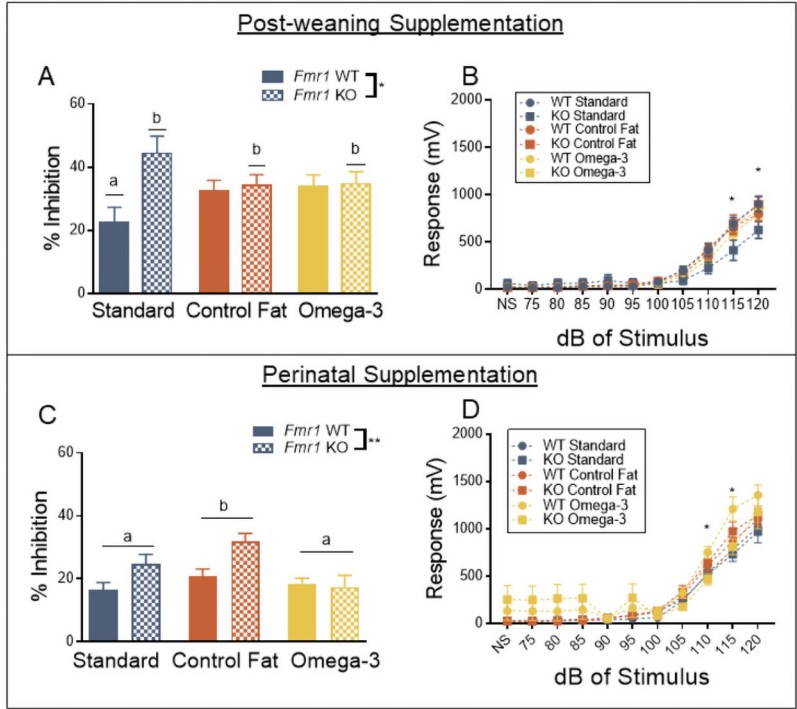

**Fig 3. Post-weaning exposure to high fat diets reversed startle threshold, while neither paradigm impacted PPI deficits.** A. Exposure to both experimental diets did not reverse exaggerated PPI in the *Fmr1* knockout in the post-weaning paradigm. B. Startle threshold was diminished in the *Fmr1* knockout, and this was reversed by exposure to both of the post-weaning experimental diets. C. Perinatal exposure to omega-3 fatty acids, as well as the control fat diet, failed to reverse PPI deficits. D. Perinatal exposure to the experimental diets did not impact the startle threshold deficits seen in the *Fmr1* knockout. Data are expressed as mean ± SEM. * = P < 0.05 ANOVA, ** = P < 0.01 ANOVA. A designation of "b" indicates that this group differed from the "a" comparison group at the level of p < 0.05 in post-hoc analyses.

are shown in the figure summarized across decibel level. Post-weaning exposure to dietary manipulation did not significantly impact percent inhibition alone, $F(2, 81) = 0.05$, $p = 0.95$, or across the levels of decibel of prepulse stimulus, $F(4, 162) = 0.15$, $p = 0.97$. Similarly, the combination of genotype and post-weaning diet was not statistically significant depending on the levels of the stimulus, $F_1(4, 162) = 0.15$, $p = 0.96$. However, overall, the combination of these two factors, genotype and diet, did in fact differentially impact percent inhibition, regardless of the level of the stimulus, $F(2, 81) = 3.70$, $p = 0.03$. Post-hoc analyses using LSD testing indicated that in standard diet condition, loss of *Fmr1* exaggerated PPI [(a) Standard WT vs (b) Standard KO, $p = 0.002$]. However, exposure to both the omega-3 [(a) Standard WT vs (b) Omega-3 KO, $p = 0.04$] and the control fat diet [(a) Standard WT vs (b) Control Fat KO, $p = 0.05$] did not differentially impact the *Fmr1* KO PPI phenotype (Fig 3A). Altogether, these results support previous findings that loss of *Fmr1* results in exaggerated prepulse inhibition, though this was unaffected by the two post-weaning dietary manipulations.

Finally, startle threshold was assessed one week later. As expected, loss of *Fmr1* did not impact the startle response overall, $F(1, 81) = 1.72$, $p = 0.19$ (Fig 3B). However, there was a significant interaction of genotype and decibel levels, $F(10, 810) = 2.88$, $p = 0.002$. Further investigation indicated that *Fmr1* KOs had reduced startle responding during this testing window at higher stimulus levels (115 and 120 dB). Post-weaning diet exposure did not impact startle responding overall, $F(2, 81) = 0.42$, $p = 0.66$, and the slope of the startle threshold curve was not shifted, $F(20, 810) = 0.98$, $p = 0.38$. The overall combination of genotype and post-weaning diet also failed to reach significance, $F(2, 81) = 0.65$, $p = 0.53$. However, the interaction of these levels with post-weaning diet and genotype did significantly impact startle responding, $F(20, 810) = 1.87$, $p = 0.01$. Post-hoc analyses indicated that at 115 dB and 120 dB, Standard KO demonstrated significantly reduced startle responding compared to Standard WT, $p = 0.05$. This was ameliorated by post-weaning exposure to both experimental post-weaning diets at both 115 dB (Control Fat KO vs Standard WT, $p = 0.97$; Omega-3 KO vs Standard WT, $p = 0.43$) and 120 dB (Control Fat KO vs Standard WT, $p = 0.94$; Omega-3 KO vs Standard WT, $p = 0.37$) (Fig 3B). Together, these results suggest that increasing the fat content of the post-weaning diet can potentially improve reduced startle responding at higher decibels in the *Fmr1* KO.

For the perinatal paradigm, loss of *Fmr1* reduced responding to the startle stimulus overall, $F(1, 84) = 5.73$, $p = 0.02$, during the habituation session (data found in S6 Fig in S1 File) and this effect become more significant over the testing window, $F(7, 588) = 2.01$, $p = 0.05$. Moreover, perinatal diet did significantly impact overall responding to the startle stimulus, $F(2, 84) = 3.49$, $p = 0.04$. Further post-hoc LSD testing indicated that the omega-3 fatty acid perinatal diet increased startle responding, compared to the standard perinatal diet, at the level of $p < 0.05$, regardless of genotype. This effect was consistent across time, $F(14, 588) = 1.59$, $p = 0.08$. The control fat perinatal diet was not significantly different from either the standard or omega-3 perinatal diet. The unique combination of genotype and perinatal diet did not significantly impact startle responding during the habituation task, both overall, $F(2, 84) = 0.35$, $p = 0.70$, or across time, $F(14, 588) = 0.54$, $p = 0.91$.

When looking at the prepulse inhibition phase, loss of *Fmr1* significantly increased percent inhibition overall, $F(1, 84) = 7.74$, $p = 0.01$ (Fig 3C). This sensitivity was also greater with increasing decibel levels, $F(2, 168) = 5.32$, $p = 0.01$. Results also indicated that perinatal diet significantly altered percent inhibition, regardless of genotype, $F(2, 84) = 5.55$, $p = 0.01$. Post-hoc analyses indicated that the control fat perinatal diet significant increased PPI compared to the standard perinatal diet and omega-3 perinatal diet, while the omega-3 perinatal diet was not significantly different from the standard perinatal diet, at the level of $p < 0.05$ (Fig 3C). This effect was not dependent on decibel levels, $F(4, 168) = 1.93$, $p = 0.11$. The combination of

genotype and perinatal diet did not impact this behavior, both overall, $F(2, 84) = 2.89$, $p = 0.06$, and across decibel levels, $F(4, 168) = 1.89$, $p = 0.11$.

For the startle threshold test, loss of *Fmr1* reduced startle responding at higher decibel levels, $F(10, 840) = 2.14$, $p = 0.02$ (Fig 3D). This was not indicative of overall lowered startle responding in this task, $F(1, 84) = 0.02$, $p = 0.88$. Perinatal diet also affected the startle threshold curve, both overall, $F(2, 84) = 4.01$, $p = 0.02$, and across the different levels, $F(20, 840) = 2.89$, $p = 0.001$. Overall, the combination of perinatal diet and genotype did not impact startle responding, $F(2, 84) = 0.28$, $p = 0.76$. However, it did alter the threshold of startle responding, $F(20, 8840) = 2.78$, $p = 0.0001$. Subsequent analyses indicated the interaction between genotype and perinatal diet was only significant at the level of 110 and 115 decibels. Further testing indicated in WT animals, the omega-3 perinatal diet increased startle responding, [Omega-3 WT vs Standard WT, $p < 0.05$; Omega-3 WT vs Standard KO, $p < 0.05$; Omega-3 KO vs Standard WT, $p < 0.05$] (Fig 3D). Moreover, perinatal control fat diet shifted both the WT and KO animals to an intermediate position that was not statistically different between either group. Together these findings suggest that neither experimental perinatal diet improved reduced startle responding following loss of *Fmr1*.

## Post-weaning supplementation with high fat diets impairs training, contextual and cued recall, while perinatal supplementation improved training in the delay fear conditioning task

In previous studies of the *Fmr1* KO, the delay fear conditioning paradigm demonstrates impaired acquisition of a fear response and impairments in cued recall [5]. Thus, we chose this paradigm to assess the ability of our treatment to improve this cognitive dysfunction. For the post-weaning paradigm, results for acquisition indicated no effect of loss of *Fmr1* on overall freezing during the acquisition of a fear memory, $F(1, 80) = 1.06$, $p = 0.31$ (Fig 4A). This effect was consistent across time, $F(4, 320) = 1.05$, $p = 0.38$. Assignment to either one of the two experimental post-weaning diets did, however, reduce freezing behavior during acquisition, $F(2, 80) = 11.38$, $p = 0.0001$. Moreover, the magnitude of this effect was different across the testing window, $F(8, 320) = 5.99$, $p = 0.0001$. In fact, beginning at tone 1, both omega-3 and control fat post-weaning diets, across genotypes, displayed reduced levels of freezing, compared to standard post-weaning diet controls, at the level of $p < 0.05$, and the magnitude of the difference grew over time. However, this effect was also dependent on genotype, as the interaction term was significant, both overall, $F(2, 80) = 3.18$, $p = 0.05$, and across time, $F(8, 320) = 2.15$, $p = 0.03$. Subsequent analyses indicated that this interaction reached significance only at the first inter-trial interval, the second presentation of the tone, and the second inter-trial interval at the level of $p < 0.05$. Analysis using a unique grouping variable indicated that at ITI 1, tone 2 and ITI 2, all other groups displayed diminished freezing compared to the Standard WT group, at the level of $p < 0.05$ (Fig 4A). Altogether, these results suggest that post-weaning exposure to both experimental post-weaning diets reduced short-term learning of the association between the tone (CS) and shock (US) in this paradigm.

The next day, conditioning to the context was measured via freezing across the 5 minute window. As expected, loss of *Fmr1* did not impact freezing behavior overall, $F(1, 80) = 1.22$, $p = 0.27$ (Fig 4B). This effect was consistent across the testing window, $F(4, 320) = 1.61$, $p = 0.17$. Post-weaning diet did significantly impact freezing behavior to the conditioned context, $F(2, 80) = 11.07$, $p = 0.0001$. Subsequent analyses with LSD post-hoc multiple comparisons indicated that animals exposed to both omega-3 ("b") and control fat post-weaning diets ("b") displayed significantly reduced freezing behavior in the conditioned context compared to the standard post-weaning diet ("a"), $p < 0.05$ (Fig 4B). This effect was consistent across the

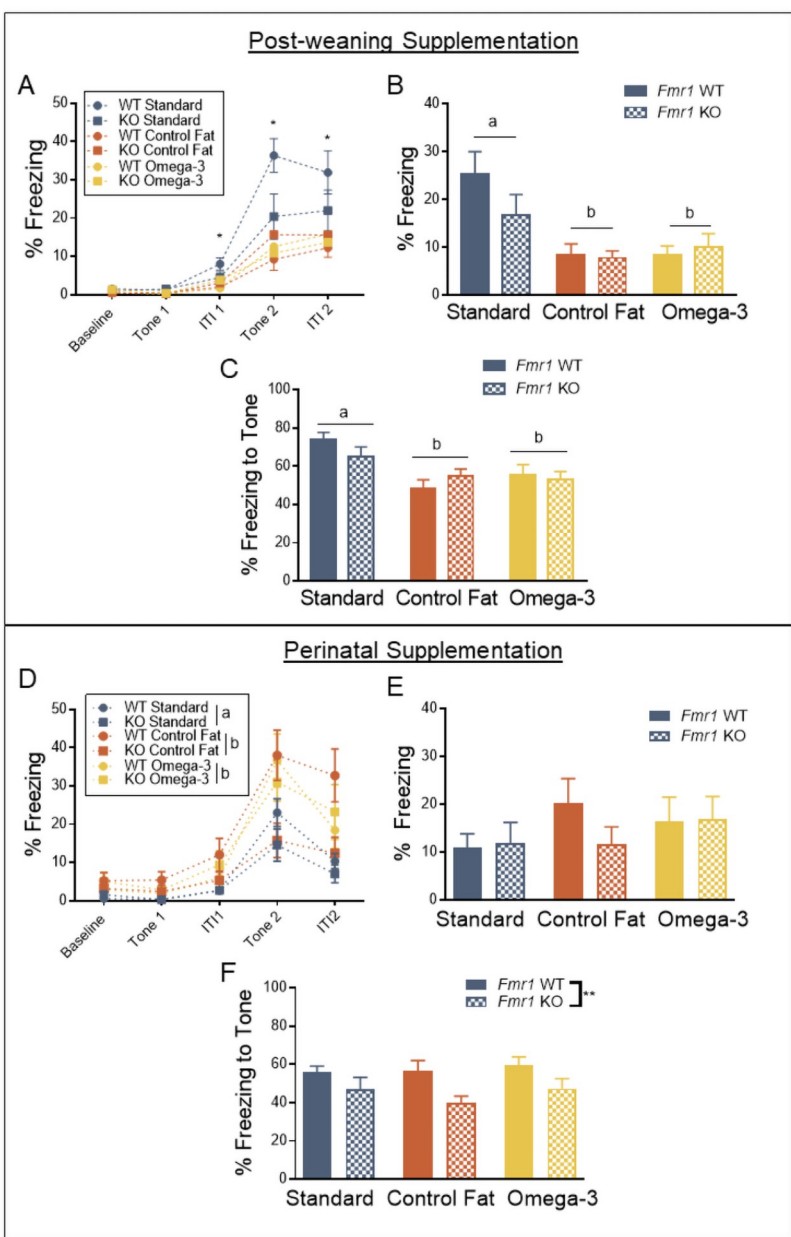

**Fig 4. Post-weaning supplementation with high fat diets impaired training, contextual and cued recall, while perinatal supplementation improved training in the delay fear conditioning task.** A. Acquisition of a fear memory was impaired by exposure to both experimental post-weaning diets in wildtype animals, and this effect was not additive for the *Fmr1* knockout. B. Contextual memory was impaired in both wildtype and knockout animals in the experimental post-weaning diet conditions. C. Cued recall was impaired by both post-weaning diets in wildtype and knockout animals, and this effect was not additive in the *Fmr1* knockout. D. Perinatal omega-3 fatty acids improve training, while the control fat perinatal diet exhibits no effect. E. No effect of either perinatal diet or genotype was detected for contextual memory. F. No effect of either perinatal diet or genotype was detected for cued recall. Data are expressed as mean ± SEM. * = $p < 0.05$ ANOVA, ** = $p < 0.01$ ANOVA. A designation of "b" indicates that this group differed from the "a" comparison group at the level of $p < 0.05$ in post-hoc analyses.

testing window, $F(8, 320) = 0.19$, $p = 0.83$. The combination of genotype and post-weaning diet did not significantly impact freezing behavior, both overall, $F(2, 80) = 1.63$, $p = 0.20$, and across time, $F(8, 320) = 1.37$, $p = 0.21$. These results suggest that post-weaning exposure to the high-fat experimental diets reduces contextual fear learning.

Following a two-hour rest, animals were placed in a novel context and the level of conditioning to the stimulus was measured via freezing to the tone expression. Loss of *Fmr1* did not alter freezing behavior across the testing window, $F(1, 80) = 1.0$, $p = 0.76$ (Fig 4C). This effect was consistent across time, $F(1, 80) = 1.46$, $p = 0.23$. However, post-weaning diet significantly impacted freezing overall during this window, $F(2, 80) = 9.88$, $p = 0.0001$. Further multiple-comparisons with LSD indicated that exposure to both omega-3 ("b") and control fat post-weaning diets ("b") significantly reduce freezing behavior overall compared to the standard post-weaning diet condition ("a") (Fig 4C). This effect was consistent across the testing window, $F(2, 80) = 1.46$, $p = 0.24$. Because the effect on freezing was the same across the testing window, only freezing in response to the cue is shown in the figure. The combination of genotype and post-weaning diet did not significantly impact freezing behavior, both overall, $F(2, 80) = 0.39$, $p = 0.68$, and across time, $F(2, 80) = 2.04$, $p = 0.14$. These results suggest that post-weaning exposure to both experimental diets reduced cued recall of the fear memory. It should be noted that based on previous findings, we expected to see a reduction in freezing to the tone based on previous findings from our group [5], and this lack of effect in the *Fmr1* knockout may be driven by the later (PD90+) testing time point.

For the perinatal paradigm, the results showed a different pattern. During the acquisition phase, loss of *Fmr1* resulted in no change overall to freezing levels across the testing window, $F(1, 90) = 3.40$, $p = 0.07$ (Fig 4D). However, the interaction of genotype and time did significantly impact freezing behavior, $F(4, 360) = 6.27$, $p = 0.001$ (Fig 4D). As expected, post-hoc multiple comparisons indicated that loss of *Fmr1* resulted in diminished freezing starting at the second iteration of the tone, at the level of $p < 0.05$. Perinatal diet significantly impacted freezing overall during this acquisition phase, $F(2, 90) = 5.08$, $p = 0.01$, as well as across time, $F(8, 360) = 3.34$, $p = 0.001$. Further multiple comparisons testing indicated that both omega-3 fatty acid ("b") and control fat ("b") perinatal diets significantly increased freezing behavior across the testing window, at the level of $p < 0.05$, and the magnitude of this effect increased over time (Fig 4D). There was also a trending interaction of genotype and perinatal diet, $F(2, 90) = 2.77$, $p = 0.07$, though this failed to reach significance. No significant three-way interaction was detected, $F(8, 360) = 1.65$, $p = 0.11$. Overall, these results suggested that perinatal exposure to both experimental diets improved acquisition of a fear response in the *Fmr1* KO.

Results for contextual fear conditioning, averaged across the 5 minute testing window, indicated no overall effect of genotype, $F(1, 90) = 0.47$, $p = 0.50$ (Fig 4E). No overall effect of perinatal diet was detected, $F(2, 90) = 0.87$, $p = 0.42$. Moreover, the different perinatal diet manipulations did not interact with genotype, $F(2, 90) = 0.77$, $p = 0.47$.

Results for examining memory for the conditioned tone indicated firstly that loss of *Fmr1* resulted in decreased freezing behavior across the test window, $F(1, 90) = 10.30$, $p = 0.002$ (Fig 4F). This reduced freezing was significantly greater in response to the tone, $F(1, 90) = 11.63$, $p = 0.001$ (Fig 4F). This decrement in freezing behavior was not, however, impacted by perinatal dietary manipulations, both overall, $F(2, 90) = 0.66$, $p = 0.52$, or across time, $F(2, 90) = 0.62$, $p = 0.54$. There was no significant between-subjects interaction of genotype and perinatal diet, $F(2, 90) = 0.60$, $p = 0.52$. No significant three-way interaction was detected, $F(2, 90) = 0.13$, $p = 0.55$. Overall, this suggests that the improvement in acquisition following perinatal exposure to these diets did not translate to cued recall the following day.

## Perinatal and post-weaning exposure to high fat diets differentially impacts hippocampal proinflammatory cytokine and BDNF expression

Among the potential mechanisms potentially at play here, omega-3 fatty acids have been shown to normalize expression of cytokine signaling [10, 19]. Following the conclusion of behavioral testing, expression of various cytokines in whole hippocampal samples was assayed. For the post-weaning paradigm, no effect of genotype or post-weaning diet was detected for: BDNF [$F(1, 29) = 0.04$, $p = 0.84$; $F(2, 29) = 0.39$, $p = 0.68$; $F(2, 29) = 1.07$, $p = 0.36$] (Fig 5A) or IL-1β [$F(1, 29) = 0.95$, $p = 0.34$; $F(2, 29) = 1.54$, $p = 0.23$; $F(2, 29) = 0.14$, $p = 0.87$] (Fig 5B). However, similar to previously published studies in our lab, IL-6 [$F(1, 29) = 4.57$, $p = 0.04$] (Fig 5C) and TNF-α [$F(1, 29) = 7.91$, $p = 0.01$] (Fig 5D) were significantly reduced following loss of *Fmr1* [47]. TNF-α was not significantly impacted by post-weaning exposure to any of the experimental post-weaning diets, either overall, $F(2, 29) = 0.40$, $p = 0.67$, or according to genotype, $F(2, 29) = 0.67$, $p = 0.52$. However, post-weaning dietary exposure to high levels of omega-3 fatty acids further reduced IL-6 expression, $F(2, 29) = 4.29$, $p = 0.02$. Post-hoc LSD multiple comparisons supported this conclusion, showing that exposure to the omega-3 post-weaning diet significantly reduced IL-6 expression relative to the standard post-weaning diet condition, $p < 0.01$. Moreover, this effect was consistent across genotypes, $F(2, 29) = 0.46$, $p = 0.63$.

For the perinatal paradigm, a total of 36 samples (n = 6 per group) were used to assess hippocampal expression of proinflammatory cytokines and BDNF. No effect of genotype was detected for BDNF, $F(1, 30) = 0.007$, $p = 0.93$ (Fig 5E). However, exposure to both high fat perinatal diets reduced hippocampal expression of BDNF, $F(2, 30) = 7.38$, $p = 0.003$. Post-hoc LSD multiple comparisons indicated that both high fat perinatal diets reduced hippocampal BDNF expression [(a) Standard Diet vs (b) Omega-3, $p = 0.004$; (a) Standard Diet vs (b) Control Fat, $p = 0.001$] (Fig 5E). This effect was consistent across the levels of genotype, $F(2, 30) = 0.33$, $p = 0.72$. A similar pattern was detected for IL-1β. No effect of genotype was detected, $F(1, 30) = 0.42$, $p = 0.52$ (Fig 5F). However, exposure to the two perinatal dietary manipulations reduced expression of IL-1β, $F(2, 30) = 8.03$, $p = 0.002$. Post-hoc LSD multiple comparison supported this [(a) Standard Diet vs (b) Omega-3 Diet, $p = 0.001$; (a) Standard Diet vs (b) Control Fat Diet, $p = 0.002$] (Fig 5F). This effect was consistent across genotypes as well, $F(2, 29) = 0.16$, $p = 0.85$. Unlike the post-weaning paradigm, no effects were seen for either IL-6 [$F(1, 30) = 0.04$, $p = 0.84$; $F(2, 30) = 2.62$, $p = 0.09$; $F(2, 29) = 0.02$, $p = 0.98$] (Fig 5G) or TNF-α [$F(1, 30) = 0.38$, $p = 0.54$; $F(2, 30) = 2.16$, $p = 0.13$; $F(2, 29) = 0.11$, $p = 0.89$] (Fig 5H). Overall, these results suggest that perinatal exposure to both dietary manipulations reduced hippocampal expression of BDNF and IL-1β regardless of expression of *Fmr1*.

## Reductions in proinflammatory cytokine and BDNF signaling are associated with fear conditioning performance

Given the nature of the experimental design used, it is difficult to determine if these changes in cytokine expression are related to the behavioral changes. Thus, we next aimed to determine if these changes were statistically associated with the behavioral phenotypes shown in the current study. For each correlation, animals from all diets were correlated across the measures of interest. These results can be found in S3 Table.

Given the decrements in acquisition seen following post-weaning administration of omega-3 fatty acids, we were most interested in whether the expression of IL-6 was associated with freezing behavior during the fear conditioning task. As expected, reductions in IL-6 expression were significantly associated with reductions in freezing behavior to the second presentation of the tone during the acquisition phase, $r(35) = 0.40$, $p = 0.02$ (Fig 6A). Reductions in IL-6

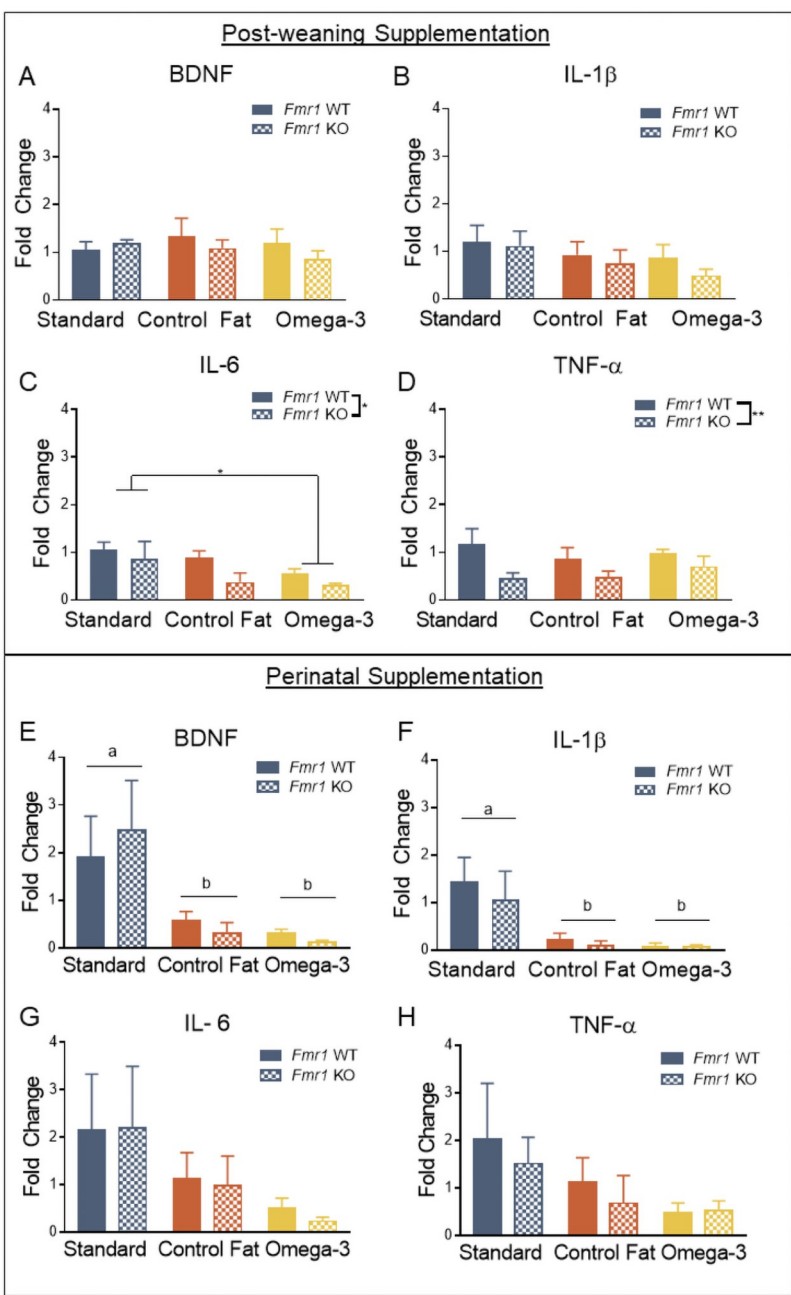

**Fig 5. Perinatal and post-weaning exposure to high fat diets differentially impacted hippocampal proinflammatory cytokine and BDNF expression.** A. Hippocampal expression of BDNF was not impacted by loss of *Fmr1* or exposure to experimental post-weaning diets. B. Hippocampal IL-1β expression was not impacted by loss of *Fmr1* or exposure to experimental post-weaning diets. C. Expression of IL-6 in the hippocampus was diminished in the *Fmr1* knockout, and exposure to omega-3 fatty acids also reduced its expression. The control fat post-weaning diet had no effect on IL-6 expression. D. TNF-α expression was also reduced in the *Fmr1* knockout with no effect of post-weaning diet. E. BDNF expression was reduced by both experimental perinatal diets. F. Similar to BDNF, IL-1β was reduced in the hippocampus of animals exposed to both experimental perinatal diets. G. Unlike the postnatal paradigm, no effect was detected for IL-6. H. No effects were detected for TNF-α. Data are expressed as mean ± SEM. * = *p* < 0.05 ANOVA, ** = *p* < 0.01 ANOVA. A designation of "b" indicates that this group differed from the "a" comparison group at the level of *p* < 0.05 in post-hoc analyses.

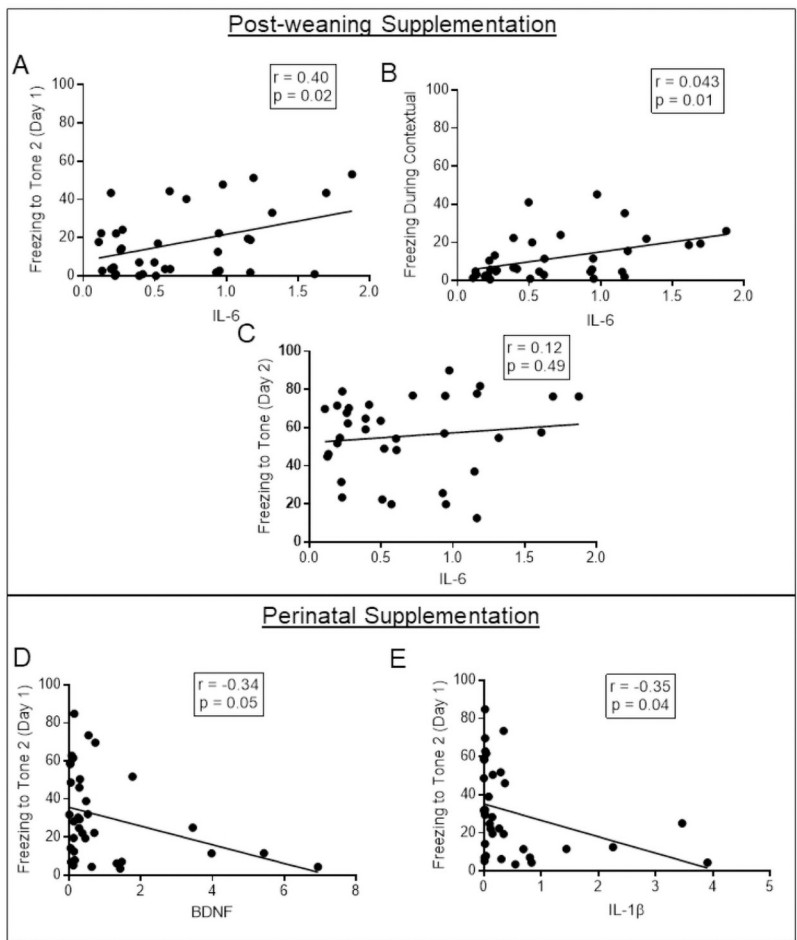

**Fig 6. Reductions in proinflammatory cytokine and BDNF signaling were associated with fear conditioning performance.** A. In the post-weaning supplementation group, reductions in hippocampal IL-6 expression were significantly associated with the reduced freezing behavior seen during the acquisition of delay fear conditioning, specifically during the second presentation of the tone. B. Similarly, reduced freezing behavior during the contextual phase was also significantly associated with IL-6 expression in the animals receiving the post-weaning intervention. C. The association between IL-6 and freezing during the cued recall phase in the post-weaning dietary intervention groups was not significant. D. Concurrent with improvements seen in acquisition of a fear memory, BDNF is significantly negatively associated with freezing during the second presentation of the tone in the cohort of animals receiving perinatal dietary manipulations. E. Similar results were found for IL-1β expression in animals receiving perinatal dietary manipulations. Data points represent individual subject scores.

expression were also significantly associated with freezing behavior during the contextual phase of the paradigm, $r(35) = 0.43$, $p = 0.01$ (Fig 6B). However, this same pattern did not hold up for cued learning the second day, $r(35) = 0.12$, $p = 0.49$ (Fig 6C). No other cytokines measured were significantly correlated with these measures, suggesting this is specific to IL-6. These results are in line with the understanding that cued learning is considered amygdala-dependent, while contextual conditioning is hippocampal-dependent, and as such we would not expect to see a relationship with hippocampal expression [40].

Given the studies showing that cytokine signaling predicts hyperactivity, we also wanted to determine if these reductions in IL-6 or other individual differences in expression levels were significantly associated with our measures of activity levels in the elevated plus maze [48]. The

only significant association was between hippocampal expression of TNFα and velocity during the task, $r(35) = -.40$, $p < 0.05$.

Similar to the post-weaning paradigm, correlations were conducted to ascertain if changes in cytokine expressions were related to any changes in behavior in animals receiving the perinatal dietary manipulations (summarized in S3 Table). Given the improvements in fear learning acquisition seen in the perinatal paradigm concurrent with reductions in BDNF and IL-1β expression, we also assessed if the expression of BDNF and IL-1β was similarly associated with freezing behavior. Indeed, results of the Pearson correlation indicated that there was a significant negative association between hippocampal BDNF expression and freezing to the second tone during the acquisition phase, $r(35) = -.34$, $p = 0.05$ (Fig 6D). Similar results were demonstrated for IL-1β expression in the hippocampus, $r(35) = -.35$, $p = 0.04$ (Fig 6E). However, these cytokine levels were not significantly related to any other time point during the fear conditioning protocol.

Similar to the post-weaning paradigm, we found it pertinent to examine whether these reductions in cytokine signaling in the hippocampus were related to changes in activity levels. However, results suggested that no cytokines measured were significantly related to either velocity or distance moved in the elevated plus maze.

## Discussion

Hyperactivity is a significant clinical component of the FXS phenotype [49–52]. Previous clinical studies demonstrated that treatment with omega-3 fatty acids during a similar window reduced hyperactivity in individuals with ASD [11]. The hyperactivity phenotype is a consistent facet of the *Fmr1* phenotype [45, 46, 53] and previous animal studies have indicated that treatment with anti-inflammatory agents also reduced hyperactivity [54]. Based on these findings, we hypothesized that dietary supplementation would ameliorate hyperactivity. Indeed, the results of the current study demonstrated that perinatal exposure to omega-3 fatty acids attenuated hyperactivity shown in the standard diet KO animals, though this was specific to perinatal supplementation (summarized in Fig 7). This effect was specific to the type of fatty acid, as the control fat diet had no impact on activity. This effect was concurrent with reductions in hippocampal BDNF expression as assessed by RT-PCR. While BDNF is typically considered a beneficial neurotrophic factor that promotes neuronal survival, in the *Fmr1*-null brain, BDNF-TrkB signaling is upregulated and loss of BDNF improves hyperactivity and altered sensorimotor gating in the *Fmr1* model [55, 56]. Altogether, the results of the present study find that perinatal supplementation with omega-3 fatty acids is sufficient to attenuate hyperactivity in the *Fmr1* knockout, which is accompanied by changes in BDNF expression in the hippocampus.

Alterations in sensorimotor gating behaviors like startle responding are a significant clinical component for children with FXS, and these changes are also seen in the *Fmr1* knockout [39, 42, 43]. Previous studies supported the possibility that this treatment would be effective, as shown in the ketamine model of schizophrenia, changes in prepulse inhibition phenotypes are amenable to treatment with omega-3 fatty acids [57]. Results from both the post-weaning and perinatal paradigm indicated that the exaggerated PPI shown in the KO mouse was not affected by exposure to either experimental diet, while reduced startle threshold was indeed ameliorated by post-weaning exposure to both omega-3 and control fat diets (summarized in Fig 7). Previous work has demonstrated that startle responding and inhibition of the startle response with a prepulse stimulus rely on separate circuitry and mechanisms, suggesting they may be differentially responsive to such treatments [58, 59]. More work is needed to determine the mechanism in how omega-3 fatty acids reversed deficits in startle responding.

**Fig 7. Summary of the main findings.** The results of the present study provide preliminary support for omega-3 fatty acids to ameliorate particular aspects of the *Fmr1* KO phenotype, including hyperactivity and sensorimotor gating deficits. ↑ = increase in behavioral output; ↓ = decrease in behavioral output. For example, ↑ under KO activity level indicates hyperactivity.

The current study finds that the timing of the intervention also mattered for the impact on fear learning and memory (summarized in Fig 7). Clinical data demonstrates mixed findings regarding the efficacy of omega-3 fatty acids on aspects of cognitive functioning [60–63]. Yet, given that in a previous study omega-3 supplementation attenuated deficits in hippocampal-dependent novel object recognition in the knockout, we expected that other aspects of nonspatial hippocampal-dependent memory might be similarly improved by this treatment [10]. Moreover, recent work also has found evidence that exposure to omega-3s can improve aspects of cognitive function following loss of *Fmr1* [22]. Conversely, the present studies found that post-weaning exposure to the two experimental diets diminished freezing behavior during acquisition, thus inhibiting learning of the acquired response. The differential acquisition of

fear conditioning presents difficulty in interpretation of the contextual conditioning and cued recall results, as when tested 24 hours later, post-weaning exposure to the experimental diets also reduced contextual fear conditioning and cued recall, regardless of genotype. The discrepancy between our findings and the impact on memory processes shown in the previous study may be due to the inclusion of the standard diet control group. In the present study, the control fat diet mirrors a typical Western diet, which has been demonstrated to have its own effects on behavior [64, 65]. Thus, the inclusion of the standard diet control renders direct comparisons between our study and the previous study difficult. However, the perinatal paradigm resulted in improved acquisition of a fear response for both diets. Previous results have indeed similarly shown that manipulation of fat content can induce large changes in developing an appropriate fear response [65–68]. Overall, the present study adds to the knowledge that manipulation of fat content across the lifespan can significantly impact fear learning and memory.

A fundamental tenet of these experiments was that loss of *Fmr1* would result in behavioral changes that mirror those previously seen. As hypothesized, we noted hyperactivity in the elevated plus maze, enhanced pre-pulse inhibition, a reduced startle threshold curve, and diminished fear learning. Moreover, no effect of loss of *Fmr1* was noted on either social behavior or stereotypy (data found in S1 File). Given what our lab [5] and others [69] have shown previously, these results in the social partition task are expected. While the present mouse model is widely considered a model for studying ASD-like characteristics, social deficits similar to clinical reports are more commonly reported in rat models with loss of *Fmr1* [22]. Moreover, the null results could be related to the testing paradigm used, given that significant social deficits previously reported were described in the three-chamber social task, while here we used the social partition task [46, 70]. The social phenotype in the present model depends on the test used [70]. Regarding the null findings for stereotypy behavior, previous work from our lab has indeed found increased rearing and stereotypy behaviors in the open field test [5], yet we did not find that result here. However, the most consistent test for detecting increased repetitive behaviors in this model has been self-grooming during social tasks and perseverance during the reversal learning portion of the Morris water maze task [71, 72]. Future studies should examine the potential impact of omega-3 fatty acids on these phenotypes to further evaluate its efficacy.

There were some other unexpected behavioral findings which may be explained by the shifts in behavioral time points across the two paradigms. For example, our results for the elevated plus maze indicated that loss of *Fmr1* resulted in decreased anxiety only in the perinatal paradigm (around the age of PD60). However, when tested after PD90 (in the post-weaning paradigm), no effect of genotype was detected. The finding from the perinatal paradigm is congruent with our previous findings, demonstrating decreased anxiety around the same time point [5]. Similarly, no impairments in cued recall were detected in the post-weaning paradigm for the delay fear conditioning task, however we did see this effect in the perinatal paradigm. The timing again lines up with the previous study conducted in our lab, and suggests that these impairments may not hold the same pattern after 3 months of age [5]. While the reason for this is unclear, this lack of effect at a later timepoint could reflect a shift to a different expression of the fear response. Many factors, including sex and individual differences, have been shown to result in divergent expression of fear responses [73, 74]. Overall, much of the work in the *Fmr1* knockout adult phenotype has focused on the early (~PD60) adult phenotype, and the results of the present study clearly indicate a need to include more time points to fully elucidate the role of *Fmr1* in fear learning.

A second fundamental hypothesis of this study was that changes in behavior resulting from these dietary manipulations could be related to changes in inflammatory cytokine signaling,

and the results of this study broadly supported this. First, our results confirmed that reduction of hippocampal IL-1β is associated with improvements in acquisition of a fear response. Next, our results expanded on previous studies from our lab, showing that post-weaning treatment with omega-3 fatty acids reduced IL-6 expression levels [9]. Prior to conducting the study, we had anticipated that a normalization of hippocampal IL-6 expression would coincide with improvements in hippocampal-dependent behaviors; however, the current study indicated that in conjunction with reduced IL-6 expression, post-weaning exposure to omega-3s was detrimental to hippocampal-dependent fear conditioning performance, but not our amygdala-dependent task. IL-6 signaling is important for many functions across the brain, including promotion of neuronal survival, protection against damage and modulation of neurotransmitter synthesis [75]. In knockout studies, IL-6 has been demonstrated to be necessary for adequate performance in the novel object recognition task, as well as water maze performance, further suggesting its importance in hippocampal function [76]. Taken as a whole, these results support that constitutive expression of IL-6, rather than overall reduction, is important for proper hippocampal functioning in the *Fmr1* model. This is in line with new evidence demonstrating that IL-6 may only take on a negative role in the presence of other proinflammatory cytokines [77].

The current study also improves our understanding of how consumption of the "Western diet" influences behavior. The control fat dietary manipulation included in the present study mirrors the typical "Western diet" [78]. Previously, it had been surmised that the high-fat diet was problematic for aspects of neurological functioning, resulting in significant deficits in learning and memory [79, 80]. Indeed, here we demonstrated that post-weaning exposure to this diet exaggerated the reduced anxiety characteristic of the *Fmr1* knockout. However, the results of the current study also complicate this thesis, finding that a post-weaning diet high in monounsaturated and saturated fats reverses deficits in startle responding. Instead of solely focusing on the negative aspects a higher fat diet, this study seems to contribute to a larger body of evidence that implicates impairments in fatty acid metabolism for the etiology of neurodevelopmental disorders [81]. Going forward, the importance of a nuanced understanding of the role of fatty acid metabolism in neural functioning is an important consideration.

Altogether, the current study provides strong evidence of omega-3 fatty acids shaping behavior in the *Fmr1* knockout mouse, further preclinical and clinical research should be conducted to determine the translational relevance. For example, the efficacy in females, the exact mechanism and the appropriate dosages all need to be determined in mouse models before establishing the potential efficacy in humans. Previously, preclinical research has produced several appealing treatment options, yet clinical trials to date have been inconclusive regarding the long-term efficacy, safety, and tolerability in human populations. Similarly, much more work is needed to determine the potential of this dietary intervention as an alternative to pharmaceuticals for individuals with FXS. However, beyond coping with the burden of the disease itself, the importance of finding affordable treatment options cannot be overstated. Nearly half of all families of children with FXS report a significant financial burden of the disease on their family, as out of pocket expenses can account for over 5% of family income [82]. Moreover, medication or therapy can account for over 50% of these out-of-pocket expenses [82]. This financial burden can disproportionately affect low-income families, as low-income families with children with special health care needs are 11 times more likely than higher income families to have out-of-pocket medical expenditures exceeding 5% of income [83]. Given the rising cost of pharmaceuticals [84], dietary interventions present as a more cost-effective alternative, further improving the lives of individuals with Fragile X syndrome.

## Supporting information

**S1 File. Additional methods and results.**
(DOCX)

**S1 Table. ANOVA results for post-weaning paradigm.**
(DOCX)

**S2 Table. ANOVA results for prenatal paradigm.**
(DOCX)

**S3 Table. Cytokine associations with behavior for both paradigms.**
(DOCX)

## Acknowledgments

We would like to acknowledge the use of equipment in the Molecular Biosciences Core.

## Author Contributions

**Conceptualization:** Suzanne O. Nolan, Joaquin N. Lugo.

**Data curation:** Matthew S. Binder, Gregory D. Smith, James T. Okoh.

**Formal analysis:** Suzanne O. Nolan, Samantha L. Hodges, Matthew S. Binder, Joaquin N. Lugo.

**Investigation:** James T. Okoh, Taylor S. Jefferson, Brianna Escobar.

**Supervision:** Joaquin N. Lugo.

**Validation:** Gregory D. Smith.

**Writing – original draft:** Suzanne O. Nolan, Samantha L. Hodges, Joaquin N. Lugo.

**Writing – review & editing:** Suzanne O. Nolan, Matthew S. Binder, Joaquin N. Lugo.

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
