## [Decision Letter · Decision Letter 0]

23 Jul 2021

PONE-D-21-20764

Dietary rescue of adult behavioral deficits in the Fmr1 knockout mouse

PLOS ONE

Dear Dr. Lugo,

Thank you for submitting your manuscript to PLOS ONE. After careful consideration, we feel that it has merit but does not fully meet PLOS ONE’s publication criteria as it currently stands. Therefore, we invite you to submit a revised version of the manuscript that addresses the points raised during the review process.

We look forward to receiving your revised manuscript.

Kind regards,

Alexandra Kavushansky, PhD

Academic Editor

PLOS ONE

Journal Requirements:

5. Please upload a copy of Figure 8, to which you refer in your text on page xx. If the figure is no longer to be included as part of the submission please remove all reference to it within the text.

6. Please include a copy of Table 8 which you refer to in your text on page 34.

Reviewers' comments:

Reviewer's Responses to Questions

**Comments to the Author**

1. Is the manuscript technically sound, and do the data support the conclusions?

Reviewer #1: Yes

Reviewer #2: Yes

2. Has the statistical analysis been performed appropriately and rigorously? 

Reviewer #1: Yes

Reviewer #2: Yes

3. Have the authors made all data underlying the findings in their manuscript fully available?

Reviewer #1: Yes

Reviewer #2: Yes

4. Is the manuscript presented in an intelligible fashion and written in standard English?

Reviewer #1: Yes

Reviewer #2: Yes

5. Review Comments to the Author

Reviewer #1: Re: PONE-D-21-20764

Nolan et al describe the effects of dietary rescue of omega-3 fatty acids at two different time points in the Fmr1 KO mouse. It is of interest, but there are some issues with the communication of the findings.

In the Introduction, first paragraph, ‘signaling in the present population’- what present population? Do the authors mean in a clinical population? In the next paragraph, the omega-3 effects in ASD could use some additional citations, including more recent ones, to reflect the mixed nature of the findings, and also could point out that intervention once the diagnosis is established might be too late for maximal impact, further justifying the prenatal administration explored in the present study. In the prenatal use of omega-3 in ASD models (refs 21-24), might also mention Matsui et al (Neuroscience 2018; 371:407-419), as it also examined prenatal vs postnatal-only omega-3. In the final paragraph, might expand the context and explain for the reader ‘and how previous effects compare when referenced to a standard dietary control’, as readers might not already be aware of the nature of the control diets in the previous work.

Methods- For Tables 1, 2, and 4, can the Standard Diet contents also be shown? The second study mentions that all tested rodents were males. The first study does not specify. Were only males used for each study? Might discuss for the reader why that is done. Also, should explain for the reader why it was decided to test study 1 mice at 90 days but study 2 mice at 60 days, particularly as this appeared to have quite an impact on the results. In the second paragraph of Delay Fear Conditioning, ‘placed in the familiar context’ on the second line- isn’t this more specific- ‘shock-associated contest’? Not just familiar?

Results- Would be helpful to find a way to indicate within Figure 2C more of the various individual significant findings within the Figure. There are several places in the Results where the text is quite unclear regarding which diet timeframe is being discussed. For the startle and PPI results, second paragraph, ‘Diet did not significantly impact percent inhibition alone’- which diet timeframe? Also please clarify ‘However, overall, the combination of these to factors differentially impacted percent inhibition’- which two factors (among genotype, diet, and level of stimulus)? Similarly, Figure 3B legend does not specify the diet timeframe. Figure 3C legend does not mention the significant finding illustrated by a and b in the Figure itself. Later, grammar ‘these interaction of these levels’. Paragraph 5 of this section it is again not specified the timing of the diet discussed for these PPI results. In the discussion of the elevated plus maze, this is the first time that the lack of effect of the Fmr1 KO on behavior is mentioned in the context of the PD90 timing- this should be mentioned with the earlier negative findings for effects of Fmr1 KO as well. It is mentioned in this section that the control fat diet resembles the typical Western diet- this probably deserves more mention in the Discussion or Introduction for its potential salience. Figure 5 legend for 5C- ‘Cued recall was impaired in wildtype and knockout animals’- do the authors mean ‘impaired by diet’ here? Figure 6 legend, again, need to be clear about the timing of the diets in the text of the legend. I could not find Table 8. Be clear what animals are being used for the cytokine and BDNF signaling for association with fear conditioning?- all diets? Also, for ‘may instead reflect baseline differences in this model’- please explain- since there is the co-factor of all of of the dietary interventions, I’m not clear how this conclusion was made. Figure 7 again needs clarity on diet timing in the text of the legend. Next to last paragraph of Results, ‘Similar to the post-weaning paradigm’- seems the end of the sentence should make some reference to this being the preweaning paradigm.

Reviewer #2: Comments to the authors

The present article aimed to elucidate the therapeutic efficacy of omega-3 fatty acids for various behavioral and neuro-immune aspects of the Fmr1 phenotype in the mouse. Overall, I find the study interesting as omega-3 supplementation and its relation with inflammation have been described, but further research is still needed to understand the precise mechanisms and therapeutic potential of the supplementation. Moreover, the timing of the treatment is essential to understand the potential of omega-3 supplementation in neurodevelopmental psychopathologies such as autism and fragile X syndrome. I appreciate the large amount of data presented (also in supplementary information) and I find the whole manuscript interesting. However, some conceptual errors and misleading information are present; moreover, in the manuscript, the authors refer to figure 8 that I cannot find in the figure lists. For this reason, I recommend major revision.

Introduction

- The introduction presents the topic clearly, and the aim of the study is supported by the appropriate background. However, when referred to the work of Pietropaolo and co-workers the authors state that "it is unknown if administration of this intervention at an earlier time-point would result in differential effects on the behavioral and neuroinflammatory phenotypes studied". On this purpose, the authors should take into account the work of Schiavi et al. 2020 (PMID: 32912100) which assess the effect of omega-3 perinatal supplementation in fmr1 mutant rats. Here, the timing of the dietary administration is similar to the “prenatal paradigm” described by the authors and thus support the authors' findings. I suggest to discuss this study also in the discussion section to corroborate the present findings.

Methods

- As dietary supplementation in the prenatal paradigm begin with parents and ends at weaning, is more appropriate to call this period "perinatal", a time interval that includes pregnancy, parturition, and lactation, differently from the “prenatal” period that refers to the interval of time before parturition.

- The authors stated that breeders were placed on one of the three experimental diets one week before pairing. I wondered if there is a specific reason for that instead of starting the dietary manipulation from gestational day 0 (the day of conception). Moreover, from the methods section is not clear how the authors calculate the gestational day 0 (plug?) reported in figure 1 (G0). This aspect should be detailed in the text. If G0 was not calculated, the authors should remove from figure 1 "G0" and specify that the dietary manipulation begins at the moment of breeding. I recommend revising the figure to avoid misleading interpretations of the prenatal/perinatal paradigm.

- From the methods section, is not clear if the authors culled the litter or not. Culling of rodent litters in the early postnatal period (generally postnatal day 1) is a standard practice in many laboratories, to reduce the litter size-induced variability in the growth and development of pups during the postnatal period and thus increase the sensitivity of the statistical analyses used to detect treatment-related effects (Agnish and Keller, 1997). Please add information about litter size. On the contrary, if litters were not culled, please specify this in the methods section.

- In the elevated plus-maze description, the parameter analyzed should be listed (Distance Moved, Velocity, and % Time in Open Arms); moreover, the method used to calculate the % Time in Open Arms should be described.

Results

- The EPM task is principally performed to evaluate anxiety in rodents and in a second time it could be used to analyze locomotor activity. Thus, devote a single paragraph to the locomotor activity detected from the EPM could be confusing, for this reason, I suggest merging the two paragraphs relative to the EPM task and describe the results concerning anxiety and the locomotor activity together.

Discussion

- In this section, the authors state that omega-3 fatty acids are a promising alternative to pharmaceuticals for individuals with FXS. This is an overstatement that, in my opinion, needs to be mitigated, considering that clinical data on this issue are still lacking.

6. PLOS authors have the option to publish the peer review history of their article (what does this mean?). If published, this will include your full peer review and any attached files.

Reviewer #1: **Yes: **David Q. Beversdorf, MD

Reviewer #2: No

---

## [Author Response · Author response to Decision Letter 0]

21 Oct 2021

We would first like to thank the reviewers for their time and input into this manuscript. We believe we have addressed their comments. We put the original comments in black text below, followed by our rebuttal that is bulleted and in blue text. We believe the manuscript is much improved through their input. 

Reviewer #1: Re: PONE-D-21-20764

Nolan et al describe the effects of dietary rescue of omega-3 fatty acids at two different time points in the Fmr1 KO mouse. It is of interest, but there are some issues with the communication of the findings.

In the Introduction, first paragraph, ‘signaling in the present population’- what present population? Do the authors mean in a clinical population? 

• We agree that this was confusing, and this has been clarified in the text to reflect that we were referring to the clinical population with FXS. 

In the next paragraph, the omega-3 effects in ASD could use some additional citations, including more recent ones, to reflect the mixed nature of the findings, and also could point out that intervention once the diagnosis is established might be too late for maximal impact, further justifying the prenatal administration explored in the present study. In the prenatal use of omega-3 in ASD models (refs 21-24), might also mention Matsui et al (Neuroscience 2018; 371:407-419), as it also examined prenatal vs postnatal-only omega-3. 

• We thank the reviewer for these thoughtful inclusions and have expanded this section to include this reference and further discussion of why the prenatal administration is critical.

In the final paragraph, might expand the context and explain for the reader ‘and how previous effects compare when referenced to a standard dietary control’, as readers might not already be aware of the nature of the control diets in the previous work.

• We agree that this should be highlighted and as such have included further context in the final paragraph of the introduction. 

Methods- For Tables 1, 2, and 4, can the Standard Diet contents also be shown? 

• Because this is a proprietary diet formulation, we were unable to get a full ingredients list from the provider, but the methods have been amended to include the specific chow being fed to the Standard Diet group. 

The second study mentions that all tested rodents were males. The first study does not specify. Were only males used for each study? Might discuss for the reader why that is done. 

• A short explanation of the reason for not including females was added to the methods section. We agree that future studies should include these, but it was not feasible for the present study. 

Also, should explain for the reader why it was decided to test study 1 mice at 90 days but study 2 mice at 60 days, particularly as this appeared to have quite an impact on the results. 

• We agree with the reviewer that on the surface it is odd that we chose different days for testing. We chose day 60 after the perinatal treatment because it was the earliest day that the mice would be considered a young adult. We chose PD 90 for the postnatal treatment that we the mice could have a longer exposure to the high omega-3 diet. The time periods were chosen to maximize the effect of the diet. 

In the second paragraph of Delay Fear Conditioning, ‘placed in the familiar context’ on the second line- isn’t this more specific- ‘shock-associated contest’? Not just familiar?

• We have adjusted this language in the methods. 

Results- Would be helpful to find a way to indicate within Figure 2C more of the various individual significant findings within the Figure.

• This figure has been updated in accordance with this comment. 

There are several places in the Results where the text is quite unclear regarding which diet timeframe is being discussed. For the startle and PPI results, second paragraph, ‘Diet did not significantly impact percent inhibition alone’- which diet timeframe? 

• We agree that this is confusing and have made adjustments throughout the manuscript to mention the specific diet timeframe being evaluated. 

Also please clarify ‘However, overall, the combination of these two factors differentially impacted percent inhibition’- which two factors (among genotype, diet, and level of stimulus)? 

• We have clarified this in the text. 

Similarly, Figure 3B legend does not specify the diet timeframe. Figure 3C legend does not mention the significant finding illustrated by a and b in the Figure itself.

• We agree that the diet timeframe was confusing and have made adjustments accordingly. The a and b designation here is used to illuminate main effects and is discussed in the text discussing this figure. 

Later, grammar ‘these interaction of these levels’. 

• This has been amended in the manuscript. 

Paragraph 5 of this section it is again not specified the timing of the diet discussed for these PPI results. 

• We agree that the diet timeframe was confusing and have made adjustments accordingly.

In the discussion of the elevated plus maze, this is the first time that the lack of effect of the Fmr1 KO on behavior is mentioned in the context of the PD90 timing- this should be mentioned with the earlier negative findings for effects of Fmr1 KO as well. 

• We have added some discussion into the results concerning the fear conditioning data where we also found an age-related discrepancy. Additionally, there is further discussion in the discussion section regarding these specific instances. 

It is mentioned in this section that the control fat diet resembles the typical Western diet- this probably deserves more mention in the Discussion or Introduction for its potential salience. 

• A paragraph was added to the end of the discussion about the potential implications of the findings here regarding the findings concerning the Western-like diet. 

Figure 5 legend for 5C- ‘Cued recall was impaired in wildtype and knockout animals’- do the authors mean ‘impaired by diet’ here? 

• We agree that this description was confusing and have made adjustments accordingly.

Figure 6 legend, again, need to be clear about the timing of the diets in the text of the legend. 

• We agree that the diet timeframe was confusing and have made adjustments accordingly.

I could not find Table 8. 

• We apologize for this omission and this has been rectified. This table was unfortunately still called Table 8 in the text but should instead refer there to Table S3!

Be clear what animals are being used for the cytokine and BDNF signaling for association with fear conditioning?- all diets? 

• We apologize for the lack of clarity – a sentence has been added to the beginning of the section of results concerning the correlations to reflect that animals from all diets were included in these analyses. 

Also, for ‘may instead reflect baseline differences in this model’- please explain- since there is the co-factor of all of the dietary interventions, I’m not clear how this conclusion was made. 

• We agree that this conclusion was erroneous and have removed this statement from the results section. 

Figure 7 again needs clarity on diet timing in the text of the legend. 

• This has been amended in the text of that legend. 

Next to last paragraph of Results, ‘Similar to the post-weaning paradigm’- seems the end of the sentence should make some reference to this being the preweaning paradigm.

• Phrasing has been added to make this clearer. 

Reviewer #2: Comments to the authors

The present article aimed to elucidate the therapeutic efficacy of omega-3 fatty acids for various behavioral and neuro-immune aspects of the Fmr1 phenotype in the mouse. Overall, I find the study interesting as omega-3 supplementation and its relation with inflammation have been described, but further research is still needed to understand the precise mechanisms and therapeutic potential of the supplementation. Moreover, the timing of the treatment is essential to understand the potential of omega-3 supplementation in neurodevelopmental psychopathologies such as autism and fragile X syndrome. I appreciate the large amount of data presented (also in supplementary information) and I find the whole manuscript interesting. However, some conceptual errors and misleading information are present; moreover, in the manuscript, the authors refer to figure 8 that I cannot find in the figure lists. For this reason, I recommend major revision.

Introduction

The introduction presents the topic clearly, and the aim of the study is supported by the appropriate background. However, when referred to the work of Pietropaolo and co-workers the authors state that "it is unknown if administration of this intervention at an earlier time-point would result in differential effects on the behavioral and neuroinflammatory phenotypes studied". On this purpose, the authors should take into account the work of Schiavi et al. 2020 (PMID: 32912100) which assess the effect of omega-3 perinatal supplementation in fmr1 mutant rats. Here, the timing of the dietary administration is similar to the “prenatal paradigm” described by the authors and thus support the authors' findings. I suggest to discuss this study also in the discussion section to corroborate the present findings.

• We agree that more references could improve readability and have as such included this reference as well as others, and further mention in the discussion. 

Methods

As dietary supplementation in the prenatal paradigm begin with parents and ends at weaning, is more appropriate to call this period "perinatal", a time interval that includes pregnancy, parturition, and lactation, differently from the “prenatal” period that refers to the interval of time before parturition.

• We agree that perinatal may be a more appropriate label for the timeframe chosen and have updated the manuscript, supplemental information and figures from prenatal to perinatal. 

The authors stated that breeders were placed on one of the three experimental diets one week before pairing. I wondered if there is a specific reason for that instead of starting the dietary manipulation from gestational day 0 (the day of conception). Moreover, from the methods section is not clear how the authors calculate the gestational day 0 (plug?) reported in figure 1 (G0). This aspect should be detailed in the text. If G0 was not calculated, the authors should remove from figure 1 "G0" and specify that the dietary manipulation begins at the moment of breeding. I recommend revising the figure to avoid misleading interpretations of the prenatal/perinatal paradigm.

• We have updated the manuscript and figure 1, removing mention of G0, to make this part clearer. As the figure legend states and text state, the experimental diets were given starting 1 week prior to breeder pairing. 

From the methods section, is not clear if the authors culled the litter or not. Culling of rodent litters in the early postnatal period (generally postnatal day 1) is a standard practice in many laboratories, to reduce the litter size-induced variability in the growth and development of pups during the postnatal period and thus increase the sensitivity of the statistical analyses used to detect treatment-related effects (Agnish and Keller, 1997). Please add information about litter size. On the contrary, if litters were not culled, please specify this in the methods section.

• We have updated the methods section to reflect that no litters were culled in either paradigm. 

In the elevated plus-maze description, the parameter analyzed should be listed (Distance Moved, Velocity, and % Time in Open Arms); moreover, the method used to calculate the % Time in Open Arms should be described.

• We have updated the methods section with further description of these parameters and how they were calculated. 

Results

The EPM task is principally performed to evaluate anxiety in rodents and in a second time it could be used to analyze locomotor activity. Thus, devote a single paragraph to the locomotor activity detected from the EPM could be confusing, for this reason, I suggest merging the two paragraphs relative to the EPM task and describe the results concerning anxiety and the locomotor activity together.

• We agree that this could be confusing and thus merged together both Figure 2 and Figure 4 as well as their respective results sections. 

Discussion

In this section, the authors state that omega-3 fatty acids are a promising alternative to pharmaceuticals for individuals with FXS. This is an overstatement that, in my opinion, needs to be mitigated, considering that clinical data on this issue are still lacking.

• We have mitigated this statement and rephrased it into further evidence of a need for more clinical research.

---

## [Decision Letter · Decision Letter 1]

21 Dec 2021

PONE-D-21-20764R1Dietary rescue of adult behavioral deficits in the Fmr1 knockout mousePLOS ONE

Dear Dr. Lugo,

Thank you for submitting your manuscript to PLOS ONE. After careful consideration, we feel that it has merit but does not fully meet PLOS ONE’s publication criteria as it currently stands. Therefore, we invite you to submit a revised version of the manuscript that addresses the points raised during the review process.

We look forward to receiving your revised manuscript.

Kind regards,

Alexandra Kavushansky, PhD

Academic Editor

PLOS ONE

Journal Requirements:

Reviewers' comments:

Reviewer's Responses to Questions

**Comments to the Author**

1. If the authors have adequately addressed your comments raised in a previous round of review and you feel that this manuscript is now acceptable for publication, you may indicate that here to bypass the “Comments to the Author” section, enter your conflict of interest statement in the “Confidential to Editor” section, and submit your "Accept" recommendation.

Reviewer #1: (No Response)

2. Is the manuscript technically sound, and do the data support the conclusions?

Reviewer #1: Yes

3. Has the statistical analysis been performed appropriately and rigorously? 

Reviewer #1: Yes

4. Have the authors made all data underlying the findings in their manuscript fully available?

Reviewer #1: Yes

5. Is the manuscript presented in an intelligible fashion and written in standard English?

Reviewer #1: Yes

6. Review Comments to the Author

Reviewer #1: Re: PONE-D-21-20764R1

Nolan et al describe the effects of dietary rescue of omega-3 fatty acids at two different time points in the Fmr1 KO mouse, which they have resubmitted. It is of interest, but there still a couple of remaining issues with the communication of the findings. They have addressed the remainder of my comments.

Methods- new text at the end of the first section- ‘Only male mice were used for the present studies, as previous work from our lab did not 129 find strong evidence of a phenotype in many of the tasks included here’- do the authors mean to add ‘…in females’ here? Otherwise, please clarify

Results- For the startle and PPI results, second paragraph, ‘Diet did not significantly impact percent inhibition alone’- which diet timeframe? This detail is still missing

Discussion- Paragraph 4- The lack of effect of the mutation on social behavior and stereotypy is fleetingly mentioned- should briefly place this in context of what is expected. Finally, the last paragraph should present more awareness of how much needs to be done before this can be translated clinically.

7. PLOS authors have the option to publish the peer review history of their article (what does this mean?). If published, this will include your full peer review and any attached files.

Reviewer #1: **Yes: **David Q. Beversdorf, MD

---

## [Author Response · Author response to Decision Letter 1]

3 Jan 2022

Reviewer #1: Re: PONE-D-21-20764R1

Nolan et al describe the effects of dietary rescue of omega-3 fatty acids at two different time points in the Fmr1 KO mouse, which they have resubmitted. It is of interest, but there still a couple of remaining issues with the communication of the findings. They have addressed the remainder of my comments.

Methods- new text at the end of the first section- ‘Only male mice were used for the present studies, as previous work from our lab did not 129 find strong evidence of a phenotype in many of the tasks included here’- do the authors mean to add ‘…in females’ here? Otherwise, please clarify.

We addressed this comment by noting that we did not find strong evidence of a phenotype in female mutants. Thank you for catching this exclusion!

Results- For the startle and PPI results, second paragraph, ‘Diet did not significantly impact percent inhibition alone’- which diet timeframe? This detail is still missing

This timeframe has been clarifying in the appropriate text. 

Discussion- Paragraph 4- The lack of effect of the mutation on social behavior and stereotypy is fleetingly mentioned- should briefly place this in context of what is expected. 

We expanded this portion of the discussion to note that based on previous findings in both our labs and others that this phenotype is not typically noted, though is more common in rat models. 

Finally, the last paragraph should present more awareness of how much needs to be done before this can be translated clinically.

We expanded this paragraph to note that though many treatments have been posited by preclinical research, their efficacy in humans is still not clear and this treatment deserves the same scrutiny despite the promising results demonstrated here.

---

## [Decision Letter · Decision Letter 2]

10 Jan 2022

Dietary rescue of adult behavioral deficits in the Fmr1 knockout mouse

PONE-D-21-20764R2

Dear Dr. Lugo,

We’re pleased to inform you that your manuscript has been judged scientifically suitable for publication and will be formally accepted for publication once it meets all outstanding technical requirements.

Kind regards,

Alexandra Kavushansky, PhD

Academic Editor

PLOS ONE

Additional Editor Comments (optional):

Reviewers' comments:

Reviewer's Responses to Questions

**Comments to the Author**

1. If the authors have adequately addressed your comments raised in a previous round of review and you feel that this manuscript is now acceptable for publication, you may indicate that here to bypass the “Comments to the Author” section, enter your conflict of interest statement in the “Confidential to Editor” section, and submit your "Accept" recommendation.

Reviewer #1: All comments have been addressed

2. Is the manuscript technically sound, and do the data support the conclusions?

Reviewer #1: Yes

3. Has the statistical analysis been performed appropriately and rigorously? 

Reviewer #1: Yes

4. Have the authors made all data underlying the findings in their manuscript fully available?

Reviewer #1: Yes

5. Is the manuscript presented in an intelligible fashion and written in standard English?

Reviewer #1: Yes

6. Review Comments to the Author

Reviewer #1: The authors have done an excellent job addressing the remaining comments. Looking forward to seeing the paper published!

7. PLOS authors have the option to publish the peer review history of their article (what does this mean?). If published, this will include your full peer review and any attached files.

Reviewer #1: **Yes: **Dr. David Q Beversdorf

---

## [Editor Report · Acceptance letter]

20 Jan 2022

PONE-D-21-20764R2 

Dietary rescue of adult behavioral deficits in the *Fmr1* knockout mouse 

Dear Dr. Lugo:

I'm pleased to inform you that your manuscript has been deemed suitable for publication in PLOS ONE. Congratulations! Your manuscript is now with our production department. 

Kind regards, 

on behalf of

Dr. Alexandra Kavushansky 

Academic Editor

PLOS ONE